# Trends in Prediabetes and Non-Alcoholic Fatty Liver Disease Associated with Abdominal Obesity among Korean Children and Adolescents: Based on the Korea National Health and Nutrition Examination Survey between 2009 and 2018

**DOI:** 10.3390/biomedicines10030584

**Published:** 2022-03-02

**Authors:** Kyungchul Song, Goeun Park, Hye Sun Lee, Myeongseob Lee, Hae In Lee, Jungmin Ahn, Eunbyoul Lee, Han Saem Choi, Junghwan Suh, Ahreum Kwon, Ho-Seong Kim, Hyun Wook Chae

**Affiliations:** 1Department of Pediatrics, Severance Children’s Hospital, Endocrine Research Institute, Yonsei University College of Medicine, Seoul 03722, Korea; endosong@yuhs.ac (K.S.); mslee8791@yuhs.ac (M.L.); haen0813@yuhs.ac (H.I.L.); anny0815@hanmail.net (J.A.); eblee.md@yuhs.ac (E.L.); hansaem6890@yuhs.ac (H.S.C.); suh30507@yuhs.ac (J.S.); armea@yuhs.ac (A.K.); kimho@yuhs.ac (H.-S.K.); 2Biostatistics Collaboration Unit, Yonsei University College of Medicine, Seoul 03722, Korea; gogoeun@yuhs.ac (G.P.); hslee1@yuhs.ac (H.S.L.); 3Department of Pediatrics, Jeju National University College of Medicine and Graduate School of Medicine, Jeju 63241, Korea; 4Department of Pediatrics, International St. Mary’s Hospital, Catholic Kwandong University, Incheon 22711, Korea

**Keywords:** prediabetic state, non-alcoholic fatty liver disease, trend, prevalence, child, adolescent

## Abstract

Investigations on the trends of prediabetes and non-alcoholic fatty liver disease (NAFLD) among children are scarce. We aimed to analyze the trends of prediabetes and NAFLD, as well as their association, among Korean children and adolescents from 2009 to 2018. This study investigated the prevalence of prediabetes, NAFLD, and abdominal obesity among 6327 children and adolescents aged 10–18 years according to age, sex, and body mass index (BMI) using a nationally representative survey. The prevalence of prediabetes, NAFLD, and abdominal obesity increased from 5.14%, 8.17%, and 5.97% respectively, in 2009 to 10.46%, 12.05%, and 10.51% respectively, in 2018. In age-specific analyses, an adverse trend in NAFLD was significant only in participants aged 16–18 years while the prevalence of prediabetes worsened significantly in all age groups. In BMI-specific analyses, the prevalence of prediabetes and NAFLD increased significantly only in participants with normal BMI. In logistic regression analysis, the odds ratio of prediabetes for NAFLD was 1.85 and those of abdominal obesity for prediabetes and NAFLD was 1.85 and 9.34, respectively. Our results demonstrated that the prevalence of prediabetes and NAFLD was increasing in association with abdominal obesity in Korean children and adolescents.

## 1. Introduction

Prediabetes is characterized by an abnormally elevated blood glucose level that is yet to be classified as diabetes mellitus but can progress to diabetes mellitus without proper management [1]. Since prediabetes is strongly associated with cardiovascular diseases and metabolic syndrome, proper screening of prediabetes is essential to manage cardiovascular diseases, especially in children and adolescents [1]. The prevalence of prediabetes among adults is high, with 33.4% in the US and 38.3% in Korea [2,3]. However, investigations on the recent trends of prediabetes among Korean children and adolescents are scarce.

Non-alcoholic fatty liver disease (NAFLD), a chronic liver disease induced by excessive fat accumulation in the liver, is a spectrum of progressive liver disease that encompasses simple steatosis, non-alcoholic steatohepatitis, advanced fibrosis, and cirrhosis [4,5]. The incidence of NAFLD is increasing globally and has become the leading cause of chronic liver disease [6], and the prevalence peaks at 70–90% among the obese and those who are diabetic [7]. The prevalence of NAFLD in the US was estimated to be 24.13% in a meta-analysis study [8]. In Korea, NAFLD prevalence was 30.7% among men and 21.6% among women in 2015–2017 [9]. However, there are limited publications in the current literature regarding the recent trends of NAFLD among Korean children and adolescents.

In addition, NAFLD is associated with risk factors of cardiovascular diseases, such as obesity, insulin resistance, and dyslipidemia [4]. NAFLD and metabolic disorders, such as prediabetes share insulin resistance as a common pathophysiological mechanism, which has led to the consideration of a more appropriate term, metabolic associated fatty liver disease, to replace NAFLD [10]. Ebe et al. [11] reported that obese adolescents with fatty liver had impaired insulin action in the liver and muscle. Insulin resistance promotes the uptake of hepatic fatty acids and triglycerides by increasing insulin, glucose, and fatty acids [12]. However, clinical studies investigating the association between prediabetes and NAFLD in children and adolescents are limited.

Therefore, this study aimed to investigate the secular trends of prediabetes and NAFLD from 2009 to 2018 among Korean children and adolescents, using the Korea National Health and Nutrition Examination Survey (KNHANES) data. Our objectives were to investigate: (1) the serial trends and recent prevalence of prediabetes and NAFLD during the 10-year study period according to age, sex, and body mass index (BMI) among Korean children and adolescents; (2) the association between prediabetes and NAFLD in children and adolescents.

## 2. Materials and Methods

### 2.1. Study Population

This cross-sectional study analyzed the data obtained between 2009 and 2018. The study design and patient inclusion flowchart are shown in Figure 1.

KNHANES is a national survey with a stratified and multistage sampling design conducted by the Korea Centers for Disease Control and Prevention, based on the National Health Promotion Act. This cross-sectional and nationally representative survey has been performed since 1998 to identify the health behaviors of the Korean population and the status of chronic diseases and to obtain data concerning food and nutrition consumption. The survey involves a two-stage stratified sampling method using sampling units and households as the first and second sampling units, respectively. Health surveys, examinations, and nutrition surveys are conducted for household members [13].

### 2.2. Study Variables

Patients’ data, such as age, sex, and anthropometric measurements, were collected. Height was measured to the nearest 0.1 cm using a portable stadiometer (range, 850–2060 mm; Seriter, Holtain Ltd., Crymych, UK), and weight was measured in the upright position to the nearest 0.1 kg using a calibrated balance beam scale (Giant 150N; HANA, Seoul, Korea). BMI was calculated as the weight divided by the height squared. Waist circumference (WC) was measured at the narrowest point between the lower borders of the rib cage and the iliac crest at the end of normal expiration. Height, weight, and BMI were presented as standard deviation score (SDS) values according to sex and age based on the 2017 Korean National Growth Charts [14]. Children were classified as normal (<85th percentile), overweight (85th to 95th percentile), or obese (≥95th percentile) according to their BMI. Abdominal obesity was defined as a WC more than the 90th percentile using the Korean waist reference data [15].

### 2.3. Laboratory Analysis

Blood samples were collected from an antecubital vein following an 8 h fast and were subsequently processed and immediately refrigerated. The serum levels of fasting glucose, total cholesterol, high-density lipoprotein cholesterol (HDL-C), and triglycerides were measured using a Hitachi Automatic Analyzer 7600/7600-210 (Hitachi, Tokyo, Japan). Low-density lipoprotein cholesterol (LDL-C) was calculated using the Friedewald formula (LDL-C = total cholesterol − [HDL-C + (triglycerides/5)]). Triglycerides/5 was used for serum samples with triglyceride values of ≤400 mg/dL, whereas it was set as missing for samples with triglyceride levels >400 mg/dL [16]. Non-HDL-C was calculated as total cholesterol minus HDL-C [17]. Serum aspartate aminotransferase (AST) and alanine aminotransferase (ALT) were measured by ultraviolet without pyridoxal-5′-phosphate method using commercially available kits (Pureauto S ALT, Daiichi Pure Chemicals, Tokyo, Japan).

The cut-off points for prediabetes were defined as fasting glucose levels between 100 and 125 mg/dL according to the American Diabetes Association guideline [18]. NAFLD was defined as elevated ALT (>26 U/L for boys and >22 U/L for girls) without hepatitis B virus or hepatitis C virus infection [19,20,21,22].

### 2.4. Statistical Analysis

The sampling weights were considered in all analyses to report representative estimates of the Korean population. All continuous variables were expressed as weighted means with standard errors, while categorical variables were expressed as weighted percentages with standard errors. We divided the participants into subgroups according to sex, age, BMI, and WC and analyzed the changes in the mean levels of WC, glucose, AST, and ALT and the proportion of participants with prediabetes, NAFLD, and abdominal obesity from 2009 to 2018 through the following years. The independent two-sample t-test and analysis of variance were used to compare the mean values of the continuous variables, and the Rao–Scott chi-squared test was used to compare categorical variables. To confirm the trend of characteristics of participants from 2009 to 2019, linear trend analysis was performed using coefficients of orthogonal polynomials in linear regression and logistic regression. Logistic regression analyses were performed with prediabetes and NAFLD as dependent variables and independent variables. In addition, BMI percentile and abdominal obesity were included in the logistic regression model as independent variables. All statistical analyses were performed using SAS version 9.4 (SAS Inc., Cary, NC, USA) for the complex survey design with clustering, stratification, and unequal weighting of the KNHANES sample. Statistical significance was set at *p* < 0.05.

## 3. Results

### 3.1. Participants’ Characteristics and Trends of Prediabetes and NAFLD

Among the total participants, the proportion of participants with prediabetes, NAFLD, and abdominal obesity was 7.71%, 9.56%, and 9.86%, respectively, and the proportion of participants who had all three of these conditions was 0.43% (Table 1). From 2009 to 2018, the proportion of prediabetes and NAFLD increased (Figure 2A,B), as well as those of generalized and abdominal obesity increased. Moreover, the proportion of participants who had prediabetes, NAFLD, and abdominal obesity increased (*p* for trend = 0.026). Similarly, the proportion of participants who had both prediabetes and NAFLD increased (*p* for trend < 0.01). The proportion of participants with prediabetes increased even in those without abdominal obesity, while the proportion of participants with NAFLD increased in those with abdominal obesity (Appendix A). Intake of protein and fat increased, while that of carbohydrate decreased (*p* for trend = 0.042, *p* for trend < 0.01, and *p* for trend = 0.028, respectively). Among the participants without abdominal obesity, intake of fat increased, while that of carbohydrate decreased (*p* for trend < 0.01 and *p* for trend = 0.026, respectively). These trends were similar in both sexes, but the adverse trend in the prevalence of abdominal obesity was more apparent in males (Appendix A).

### 3.2. Trends of Prediabetes and NAFLD according to Age

In age-specific analyses, adverse trends in generalized and abdominal obesity were more apparent in participants aged 16–18 years than in those aged 10–15 years (Table 2). The prevalence of prediabetes increased in all groups (All *p* for trend < 0.01), while that of NAFLD increased significantly only in participants aged 16–18 years (*p* for trend = 0.026). These trends were similar in both sexes (Appendix A).

### 3.3. Trends of Prediabetes and NAFLD according to BMI

In BMI-specific analyses, the prevalence of abdominal obesity increased significantly only in the obese group (*p* for trend = 0.024), while that of prediabetes and NAFLD increased only in participants with normal BMI (*p* for trend < 0.01 and *p* for trend = 0.019, respectively) (Table 3). In BMI- and sex-specific analyses, the adverse trend of abdominal obesity was only significant in males with obesity (*p* for trend < 0.01) (Appendix A). The prevalence of prediabetes increased in both sexes with normal BMI (*p* for trend < 0.01 for males and *p* for trend = 0.026 for females), while that of NAFLD increased in males with normal BMI (*p* for trend < 0.01).

### 3.4. Association between Prediabetes and NAFLD

In logistic regression analyses, the overweight population had 0.42 and 3.67 more odds of being prediabetic and NAFLD, respectively, compared to the normal BMI population. Odds ratios (ORs) of generalized and abdominal obesity for prediabetes were 1.88 (95% CI, 1.41–2.52; *p* < 0.01) and 1.85 (95% CI, 1.39–2.47; *p* < 0.01), and the corresponding values for NAFLD were 13.89 (95% CI, 10.91–17.66; *p* < 0.01) and 9.34 (95% CI, 7.39–11.81; *p* < 0.01), respectively (Table 4). The participants with prediabetes had 0.85 more odds of being NAFLD compared to those with normal glucose levels (95% CI, 1.37–2.52; *p* < 0.01).

Among the participants with NAFLD, the prevalence of prediabetes was 12.6%, while the corresponding value was 7.2% among those without NAFLD (Figure 3A). The prevalence of NAFLD was 15.6% among the participants with prediabetes, while the corresponding value was 9.1% among those without prediabetes (Figure 3B).

## 4. Discussion

This study demonstrated an adverse trend in generalized and abdominal obesity, prediabetes, and NAFLD among Korean children and adolescents. The prevalence of prediabetes worsened significantly in all age groups, while the adverse trends in NAFLD and abdominal obesity were more apparent in the older age group than those in the young age group. In addition, adverse trends in prediabetes and NAFLD were more prominent in participants with normal BMI than in those with obesity. In logistic regression analysis and *t*-test, prediabetes was positively associated with NAFLD. Furthermore, generalized and abdominal obesity was positively associated with prediabetes and NAFLD.

The prevalence of prediabetes, as defined by glucose abnormalities, demonstrated no significant trend from 1999 to 2014, and the total prevalence was 15.5% among the US adolescents [1]. In Iran, the corresponding value increased from 7% in 1999–2005 to 16.6% in 2011–2014 among adolescents [23]. In India, the prevalence of prediabetes was 12.3% and 8.4% among boys and girls, respectively [24]. In our study, the prevalence of prediabetes increased from 5.1% in 2009 to 10.5% in 2018, which was lower than that in the US and Iran but similar to that in India.

The prevalence of NAFLD among children and adolescents increased from 19.34 million in 1990 to 29.49 million in 2017 as per the Global Burden of Disease database-based study [25]. In a global meta-analysis, the prevalence of NAFLD was 7.6% among children and adolescents and 34.2% among obese children and adolescents [26]. In the US, the prevalence of NAFLD among adolescents increased from 3.9% in 1988–1994 to 10.7% in 2007–2010 [27]. In China, the prevalence of NAFLD was 9.03% among children [20]. In our study, the prevalence of NAFLD increased from 8.17% in 2009 to 12.05% in 2018.

The global prevalence of obesity increased from 0.9% to 7.8% in boys and from 0.7% to 5.6% in girls, from 1975 to 2016, in a pooled analysis of population-based studies [28]. A systematic review reported that the prevalence of abdominal obesity increased from 16.3% in 1985–1999 to 33.9% in 2010–2014, among the population aged 15–40 years [29]. In the US, the prevalence of obesity was 21.8% in adolescents, while that of abdominal obesity was 31.5% in boys and 38.2% in girls [30]. In China, the prevalence of obesity was 15.7%, while that of abdominal obesity was 31.4% among children and adolescents [31]. In our study, the prevalence of obesity in 2018 was 11.64%, which was higher than that in the global study but lower than that in the US and China studies. Furthermore, the prevalence of abdominal obesity was lower in our study than that in the US and China studies.

The guideline of the American Diabetes Association recommends prediabetes screening of children who are overweight or obese and have one or more risk factors for diabetes [32]. The guideline of the North American Society of Pediatric Gastroenterology, Hepatology and Nutrition suggests that screening of NAFLD should be considered in children who are obese or overweight and have additional risk factors [4]. As these guidelines do not suggest screening of children with normal BMI or consider WC, prediabetes and NAFLD in children and adolescents with normal BMI but with abdominal obesity could be under-recognized. As adverse trends in prediabetes and NAFLD were more prominent among children and adolescents with normal BMI and abdominal obesity, and abdominal obesity was positively associated with prediabetes and NAFLD in our study, more attentive evaluation and management of prediabetes and NAFLD are required in children and adolescents with normal BMI, especially in those with abdominal obesity.

NAFLD is related to a higher prevalence of diabetes and cardiovascular diseases. Moreover, diabetes is a risk factor for the development of liver fibrosis and cirrhosis [5,33]. In a cross-sectional study, children with NAFLD had nearly twice the prevalence of prediabetes or diabetes as those without [5]. Hudson et al. [34] reported that 48% of the children with type 2 diabetes had high ALT levels. In our study, children and adolescents with prediabetes had nearly twice the prevalence of NAFLD as those without, and children and adolescents with NAFLD had nearly twice the prevalence of prediabetes as those without. Therefore, close monitoring of glucose levels in children and adolescents with NAFLD as well as an attentive screening of NAFLD in those with prediabetes are required to reduce future risks of cardiovascular disease.

The adverse trend of prediabetes and NAFLD in this study might be due to the increasing prevalence of generalized and abdominal obesity associated with lifestyle changes among Korean children and adolescents. Abdominal obesity is related to cardiometabolic risk factors and NAFLD [20,35]. Our previous studies suggested that abdominal obesity might contribute to an adverse trend in metabolic syndrome and dyslipidemia [36,37].

The increased prevalence of participants with prediabetes and NAFLD in children and adolescents without abdominal obesity may be related to an unhealthy diet [38,39]. High calories, carbohydrates, and fat increase the prevalence of chronic diseases associated with insulin resistance, such as NAFLD [25,38,40]. The increased consumption of sweetened beverages, particularly among children and adolescents, leads to increased fructose adsorption with increased lipogenesis, triglycerides accumulation, and liver damage due to its metabolism [41]. Moreover, macronutrient composition in the diet can impact the incidence of these chronic diseases without affecting body weight [38]. Rhee et al. [42] reported that rapid change in dietary patterns, including an increase in energy intake with a high-fat diet, might contribute to the increasing prevalence of diabetes in Asia. Ha et al. [43] reported that carbohydrate intake was higher than the recommended range, which was 87% in men and 91% in women in Korea, and was associated with an increased incidence of type 2 diabetes. Our previous study reported that daily calorie and fat intake increased with an increase in impaired fasting glucose among Korean children and adolescents [36].

This study has some limitations. First, this was a cross-sectional study limited to the South Korean population. Second, confounding factors, such as physical activity and pubertal status, were not considered. Third, defining prediabetes according to the HbA1c was not considered in this study. Fourth, ultrasonography and liver biopsy were not performed to diagnose NAFLD because KNHANES does not include information on ultrasonography or liver biopsy. However, the study assessed the trends in the prevalence of prediabetes and NAFLD among Korean children and adolescents according to age, sex, and BMI using nationally representative data.

## 5. Conclusions

Our study demonstrated the adverse trends of prediabetes and NAFLD, as well as their relationship, among Korean children and adolescents. Moreover, the adverse trend was more apparent in children and adolescents with normal BMI and abdominal obesity, even in the young age group. These findings suggest that future risks for cardiovascular and chronic liver diseases may increase. Therefore, close monitoring of levels of fasting glucose, ALT, and WC is required not only for children and adolescents who are overweight or obese but also for those with normal BMI.

## Figures and Tables

**Figure 1 biomedicines-10-00584-f001:**
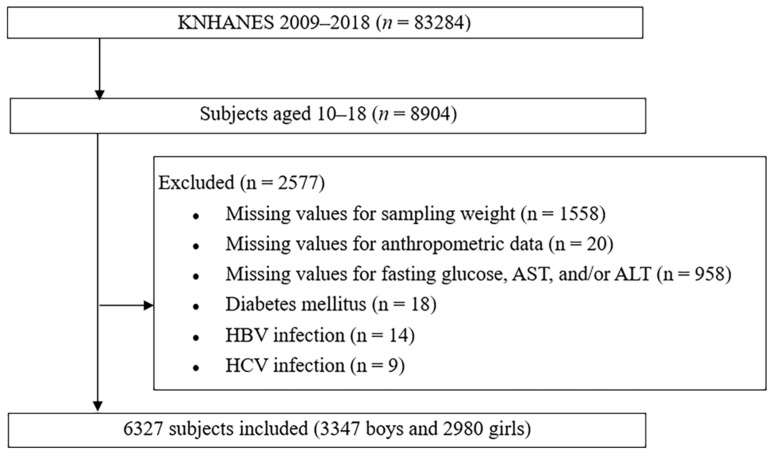
Design and flowchart of study population. KNHANES: Korea National Health and Nutrition Examination Survey; AST: aspartate aminotransferase; ALT: alanine aminotransferase.

**Figure 2 biomedicines-10-00584-f002:**
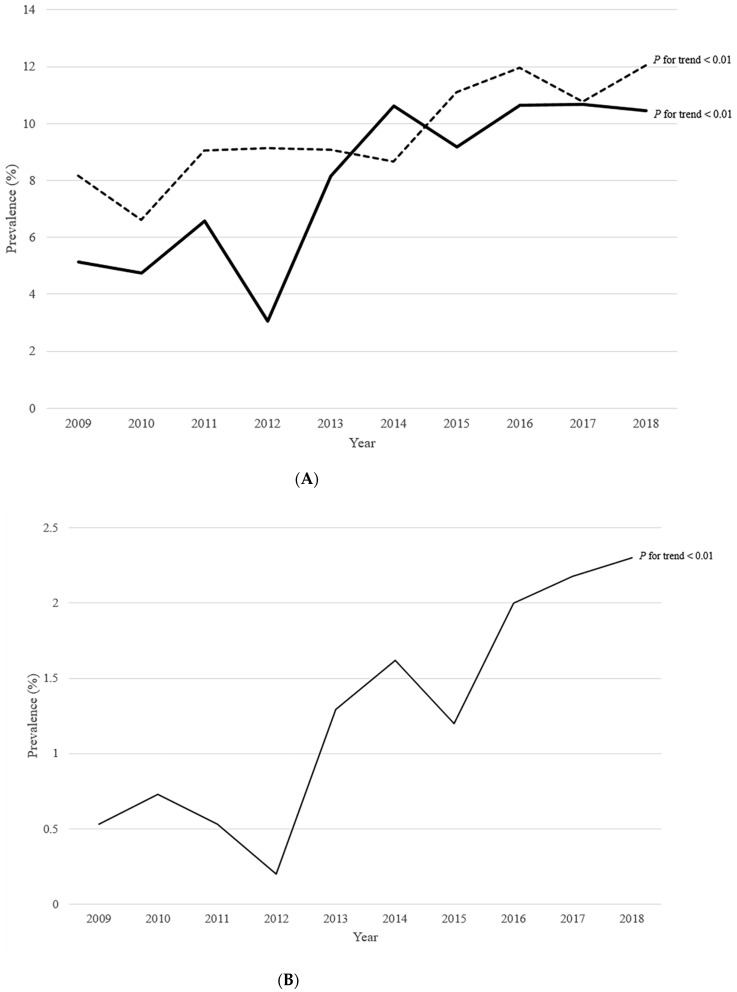
Trends of the prevalence of prediabetes and NAFLD among Korean children and adolescents between 2009 and 2018. Linear trend analysis was performed using coefficients of orthogonal polynomials to calculate *p* for trend. (**A**) The black solid line is the prevalence of prediabetes and the dashed line is the prevalence of NAFLD (**B**) The black narrow line is the prevalence of subjects with prediabetes and NAFLD. NAFLD: non-alcoholic fatty liver disease.

**Figure 3 biomedicines-10-00584-f003:**
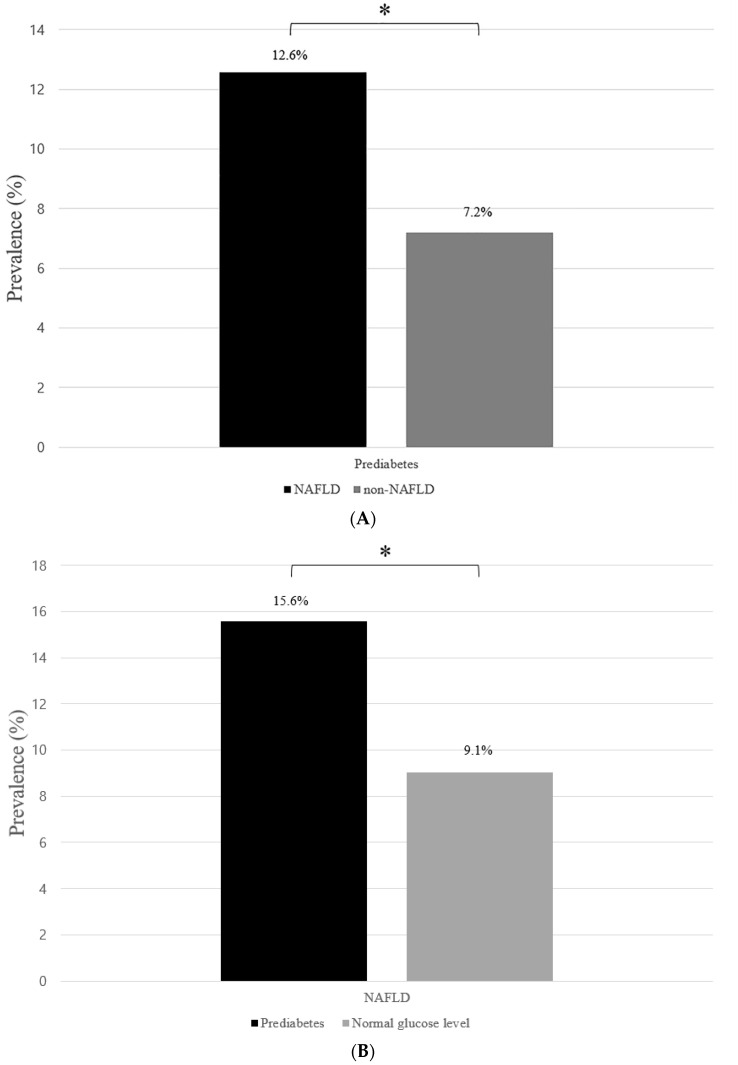
Prevalence of prediabetes according to NAFLD and those of NAFLD according to prediabetes. (**A**) Prevalence of participants with prediabetes among those with NAFLD and non-NAFLD. The “black” bar represents the NAFLD group, whereas the “gray” bar represents the non-NAFLD group. (**B**) Prevalence of participants with NAFLD among those with prediabetes and normal glucose levels. The “black” bar represents the prediabetes group, whereas the “gray” bar represents the normal glucose group. The asterisk indicates the *p*-value of the independent two-sample *t*-test < 0.01. NAFLD: non-alcoholic fatty liver disease.

**Table 1 biomedicines-10-00584-t001:** Trend of characteristics of participants.

Variable	2009	2010	2011	2012	2013	2014	2015	2016	2017	2018	Total	*p*	*p* for Trend
Total number	*n* = 965	*n* = 771	*n* = 699	*n* = 643	*n* = 672	*n* = 466	*n* = 535	*n* = 569	*n* = 517	*n* = 490	*n* = 6437		
Height SDS	0.29 (0.05)	0.25 (0.05)	0.16 (0.05)	0.28 (0.06)	0.25 (0.04)	0.17 (0.07)	0.17 (0.05)	0.24 (0.06)	0.28 (0.06)	0.30 (0.06)	0.24 (0.02)	0.509	0.845
Weight SDS	0.03 (0.05)	0.04 (0.06)	0.02 (0.05)	0.07 (0.06)	0.13 (0.05)	0.11 (0.08)	0.25 (0.07)	0.04 (0.07)	0.13 (0.06)	0.17 (0.06)	0.09 (0.02)	0.193	**0.022**
BMI SDS	−0.14 (0.05)	−0.10 (0.06)	−0.09 (0.06)	−0.09 (0.06)	0.01 (0.06)	0.03 (0.08)	0.20 (0.07)	−0.09 (0.07)	−0.01 (0.07)	0.02 (0.07)	−0.03 (0.02)	**0.015**	**<0.01**
*BMI percentile*												0.146	**<0.01**
*Normal, %*	83.53 (1.32)	81.37 (1.70)	79.92 (1.62)	81.15 (1.90)	81.26 (1.66)	80.32 (2.07)	76.74 (2.05)	78.77 (1.84)	80.56 (1.95)	80.49 (1.98)	80.50 (0.57)		
*Overweight, %*	9.92 (1.02)	10.07 (1.33)	8.39 (1.29)	9.45 (1.30)	10.02 (1.19)	8.94 (1.37)	9.02 (1.35)	9.22 (1.36)	8.33 (1.36)	7.87 (1.29)	9.15 (0.41)		
*Obesity, %*	6.55 (0.97)	8.56 (1.25)	11.69 (1.42)	9.40 (1.50)	8.73 (1.15)	10.73 (1.70)	14.24 (1.71)	12.00 (1.58)	11.12 (1.55)	11.64 (1.73)	10.35 (0.46)		
WC, cm	83.53 (1.32)	81.37 (1.70)	79.92 (1.62)	81.15 (1.90)	81.26 (1.66)	80.32 (2.07)	76.74 (2.05)	78.77 (1.84)	80.56 (1.95)	80.49 (1.98)	69.93 (0.16)	**<0.01**	**<0.01**
*Abdominal obesity, %*	5.97 (0.84)	8.13 (1.27)	10.96 (1.31)	8.18 (1.30)	8.13 (1.19)	10.23 (1.66)	14.47 (1.61)	13.07 (1.63)	10.26 (1.48)	10.51 (1.53)	9.86 (0.44)	**<0.01**	**<0.01**
Glucose, mg/dL	88.87 (0.28)	88.78 (0.29)	88.61 (0.36)	88.86 (0.40)	90.44 (0.36)	91.40 (0.39)	91.40 (0.37)	91.45 (0.30)	91.10 (0.37)	91.56 (0.36)	90.13 (0.12)	**<0.01**	**<0.01**
*Prediabetes, %*	5.14 (0.99)	4.73 (0.86)	6.56 (1.32)	3.05 (0.70)	8.15 (1.37)	10.62 (1.55)	9.18 (1.42)	10.65 (1.46)	10.68 (1.73)	10.46 (1.38)	7.71 (0.42)	**<0.01**	**<0.01**
AST, IU/L	18.68(0.19)	18.82(0.28)	18.99(0.29)	19.53(0.48)	18.41(0.26)	18.71(0.36)	19.99(0.65)	18.82(0.35)	19.87(0.53)	20.34(0.45)	19.18 (0.12)	**<0.01**	**<0.01**
ALT, IU/L	14.45(0.39)	14.51(0.53)	15.57(0.81)	15.24(0.79)	15.06(0.51)	14.64(0.65)	16.47(1.22)	15.06(0.76)	16.68(1.36)	17.31(1.18)	15.44 (0.27)	0.309	0.015
*NAFLD, %*	8.17 (1.03)	6.62 (1.16)	9.06 (1.24)	9.14 (1.58)	9.09 (1.14)	8.67 (1.80)	11.11 (1.51)	11.97 (1.89)	10.77 (1.51)	12.05 (1.74)	9.56 (0.46)	0.163	**<0.01**
*Prediabetes and NAFLD, %*	0.53 (0.23)	0.73 (0.29)	0.53 (0.33)	0.20 (0.15)	1.29 (0.47)	1.62 (0.67)	1.20 (0.62)	2.00 (0.74)	2.18 (0.77)	2.30 (0.72)	1.20 (0.16)	**0.012**	**<0.01**
*Prediabetes and abdominal obesity, %*	0.75 (0.33)	1.02 (0.45)	0.92 (0.43)	0.40 (0.23)	0.79 (0.36)	1.10 (0.49)	2.57 (0.81)	2.11 (0.64)	1.47 (0.58)	1.64 (0.59)	1.24 (0.16)	0.058	**<0.01**
*NAFLD and abdominal obesity, %*	1.90 (0.46)	1.67 (0.46)	4.68 (0.96)	2.74 (0.90)	3.62 (0.84)	3.31 (1.21)	5.28 (1.02)	4.57 (1.01)	5.30 (1.16)	6.08 (1.26)	3.83 (0.29)	**<0.01**	**<0.01**
*Prediabetes, NAFLD, and abdominal obesity, %*	0.25 (0.18)	0.36 (0.22)	0.15 (0.15)	0.13 (0.13)	0.35 (0.26)	0.48 (0.33)	0.55 (0.50)	0.42 (0.30)	0.61 (0.38)	1.18 (0.54)	0.43 (0.10)	0.452	**0.026**
Total cholesterol, mg/dL	157.73 (1.40)	156.71 (1.15)	157.85 (1.52)	159.24 (1.55)	157.47 (1.12)	158.06 (1.31)	160.44 (1.28)	163.80 (1.39)	166.89 (1.44)	165.35 (1.29)	160.14 (0.44)	**<0.01**	**<0.01**
HDL-C, mg/dL	49.32 (0.40)	49.40 (0.42)	51.41 (0.53)	51.60 (0.66)	52.16 (0.43)	51.97 (0.52)	51.34 (0.49)	51.99 (0.58)	51.83 (0.50)	51.23 (0.55)	51.17 (0.16)	**<0.01**	**<0.01**
LDL-C, mg/dL	90.70 (1.16)	91.14 (1.01)	92.06 (1.37)	91.18 (1.19)	89.43 (0.92)	89.66 (1.19)	94.19 (1.25)	95.06 (1.11)	98.44 (1.17)	97.03 (1.24)	92.74 (0.38)	**<0.01**	**<0.01**
Triglycerides, mg/dL	88.46 (2.25)	85.06 (2.77)	81.24 (2.18)	86.49 (3.12)	82.56 (2.12)	85.52 (3.00)	87.26 (2.98)	86.25 (2.41)	86.07 (3.01)	90.20 (2.63)	85.81 (0.84)	0.259	0.238
Energy, kcal/day	1971.51 (30.94)	2196.86 (45.63)	2176.26 (38.37)	2169.01 (45.05)	2171.92 (45.98)	2235.74 (56.66)	2225.05 (47.68)	2140.75 (39.92)	2102.38 (46.73)	2135.72 (53.82)	2149.92 (14.16)	**<0.01**	0.268
Carbohydrate, g/day	312.74 (5.08)	335.53 (6.63)	335.61 (5.82)	336.22 (6.31)	325.66 (6.05)	329.65 (8.10)	327.94 (6.36)	318.81 (6.04)	311.80 (6.78)	313.99 (7.97)	325.12 (2.05)	**<0.01**	**0.028**
Protein, g/day	68.73 (1.10)	77.65 (2.16)	79.87 (2.25)	80.36 (2.40)	77.94 (2.58)	80.62 (2.60)	80.43 (2.44)	80.03 (1.89)	76.72 (2.15)	77.51 (2.41)	77.86 (0.70)	**<0.01**	**0.042**
Fat, g/day	49.56 (1.31)	59.77 (2.01)	56.76 (1.58)	56.67 (1.99)	58.38 (1.88)	62.10 (2.25)	62.05 (2.09)	57.71 (1.74)	58.12 (1.91)	59.96 (2.08)	57.88 (0.59)	**<0.01**	**<0.01**

Continuous variables are presented as the mean (standard error) and categorical data as the percentage (standard error) *with Italic.* Statistically significant *p* value is presented in bold type. SDS: standard deviation score; BMI: body mass index; WC: waist circumference; AST: aspartate aminotransferase; ALT: alanine aminotransferase; NAFLD: non-alcoholic fatty liver disease; HDL-C: high-density lipoprotein cholesterol; LDL-C: low-density lipoprotein cholesterol; TyG index: triglyceride-glucose index.

**Table 2 biomedicines-10-00584-t002:** Trend of prediabetes of participants according to age classification.

Variable	2009	2010	2011	2012	2013	2014	2015	2016	2017	2018	Total	*p*	*p* for Trend
10–12 y	*n* = 340	*n* = 293	*n* = 270	*n* = 246	*n* = 239	*n* = 142	*n* = 183	*n* = 183	*n* = 190	*n* = 189	*n* = 2275		
BMI SDS	−0.12 (0.06)	−0.00 (0.09)	−0.20 (0.09)	−0.13 (0.08)	0.01 (0.09)	−0.10 (0.12)	0.17 (0.11)	−0.04 (0.12)	−0.16 (0.10)	−0.12 (0.08)	−0.07 (0.03)	0.303	0.694
*BMI percentile*												0.137	0.352
*Normal, %*	84.31 (2.09)	78.05 (2.88)	84.61 (2.75)	86.36 (2.45)	81.98 (2.70)	85.80 (3.09)	77.84 (3.59)	78.42 (3.73)	85.20 (2.99)	81.99 (2.63)	82.56 (0.91)		
*Overweight, %*	10.71 (1.94)	13.91 (2.45)	8.68 (2.13)	5.00 (1.28)	12.03 (2.20)	7.61 (2.25)	9.08 (2.27)	10.17 (2.51)	8.72 (2.57)	10.66 (2.18)	9.69 (0.70)		
*Obesity, %*	4.98 (1.16)	8.04 (1.94)	6.72 (1.73)	8.64 (2.22)	5.99 (1.64)	6.59 (2.38)	13.08 (2.75)	11.42 (2.75)	6.07 (2.20)	7.35 (2.07)	7.75 (0.66)		
WC, cm	64.95 (0.49)	65.71 (0.63)	64.34 (0.67)	64.94 (0.66)	65.33 (0.67)	65.14 (0.90)	68.70 (0.99)	66.75 (0.90)	64.56 (0.70)	65.23 (0.69)	65.49 (0.23)	**0.026**	0.108
*Abdominal obesity, %*	6.60 (1.46)	9.32 (2.13)	7.98 (1.86)	7.58 (2.10)	5.84 (1.58)	7.56 (2.43)	15.72 (3.24)	13.19 (2.87)	4.76 (1.79)	8.06 (2.05)	8.52 (0.69)	**0.017**	0.358
Glucose, mg/dL	90.53 (0.32)	90.54 (0.50)	89.78 (0.42)	90.55 (0.43)	92.67 (0.50)	93.27 (0.59)	93.00 (0.49)	93.24 (0.49)	93.40 (0.59)	93.91 (0.50)	91.91 (0.16)	**<0.01**	**<0.01**
*Prediabetes, %*	6.89 (1.81)	7.34 (1.75)	7.56 (2.13)	5.11 (1.67)	11.41 (2.68)	12.75 (2.68)	13.20 (2.69)	12.92 (2.69)	15.04 (3.22)	17.55 (2.68)	10.57 (0.77)	**<0.01**	**<0.01**
AST, IU/L	21.32 (0.43)	21.69 (0.39)	21.68 (0.50)	21.70 (1.11)	20.88 (0.49)	21.48 (0.56)	24.36 (2.06)	20.99 (0.37)	21.50 (0.48)	23.28 (0.63)	21.85 (0.26)	0.123	0.059
ALT, IU/L	15.81 (1.02)	16.16 (0.87)	16.46 (1.33)	15.16 (1.52)	15.56 (0.97)	14.51 (0.98)	19.68 (3.90)	13.92 (0.59)	14.45 (0.91)	17.13 (1.03)	15.89 (0.48)	0.152	0.993
*NAFLD, %*	8.85 (1.68)	7.67 (1.76)	7.14 (1.66)	6.92 (3.42)	7.03 (1.69)	4.64 (1.99)	12.61 (2.45)	9.10 (2.44)	6.71 (1.99)	13.83 (2.64)	8.37 (0.70)	0.197	0.132
13–15 y	*n* = 359	*n* = 295	*n* = 242	*n* = 232	*n* = 234	*n* = 180	*n* = 185	*n* = 218	*n* = 169	*n* = 160	*n* = 2274		
BMI SDS	−0.12 (0.07)	−0.13 (0.08)	0.05 (0.10)	−0.05 (0.10)	−0.07 (0.10)	−0.02 (0.10)	0.24 (0.10)	−0.05 (0.10)	−0.01 (0.10)	0.04 (0.10)	−0.02 (0.03)	0.204	0.092
*BMI percentile*												**<0.01**	0.226
*Normal, %*	81.43 (2.07)	83.26 (2.40)	74.47 (3.21)	77.75 (3.87)	85.83 (2.47)	80.54 (3.07)	77.10 (3.55)	77.76 (3.39)	79.95 (3.06)	82.54 (3.25)	80.05 (0.97)		
*Overweight, %*	10.70 (1.64)	10.78 (1.97)	9.31 (2.25)	12.72 (2.91)	7.24 (1.70)	10.57 (2.33)	9.70 (2.49)	10.26 (2.93)	8.96 (2.23)	1.39 (0.76)	9.39 (0.72)		
*Obesity, %*	7.87 (1.75)	5.95 (1.51)	16.22 (2.70)	9.52 (2.79)	6.93 (1.88)	8.89 (2.15)	13.20 (2.93)	11.98 (2.45)	11.09 (2.39)	16.07 (3.21)	10.55 (0.76)		
WC, cm	69.85 (0.44)	69.34 (0.73)	70.91 (0.71)	70.31 (0.79)	69.74 (0.83)	70.79 (0.77)	72.97 (0.76)	70.72 (0.73)	70.41 (0.79)	70.80 (0.76)	70.52 (0.23)	0.053	**0.042**
Abdominal obesity, %	4.96 (1.24)	6.29 (1.77)	12.07 (2.46)	7.59 (2.33)	9.21 (2.29)	7.56 (2.12)	11.87 (2.69)	9.99 (2.24)	10.61 (2.36)	11.25 (2.55)	8.94 (0.70)	0.209	**0.016**
Glucose, mg/dL	89.24 (0.36)	88.66 (0.44)	89.40 (0.52)	89.56 (0.61)	90.61 (0.58)	91.66 (0.59)	91.44 (0.46)	91.52 (0.50)	91.37 (0.55)	91.22 (0.64)	90.33 (0.17)	**<0.01**	**<0.01**
*Prediabetes, %*	5.28 (1.36)	4.52 (1.46)	8.34 (2.54)	3.92 (1.18)	8.68 (2.32)	11.72 (2.50)	8.33 (2.08)	10.13 (2.49)	10.65 (2.83)	10.08 (2.40)	7.90 (0.68)	0.074	**<0.01**
AST, IU/L	17.95 (0.25)	17.71 (0.31)	17.66 (0.36)	18.37 (0.63)	17.33 (0.34)	17.96 (0.48)	18.46 (0.40)	18.23 (0.41)	20.96 (1.57)	19.18 (0.53)	18.29 (0.19)	0.063	**<0.01**
ALT, IU/L	13.58 (0.46)	12.73 (0.59)	13.69 (0.76)	14.35 (1.57)	12.95 (0.60)	13.27 (0.81)	13.96 (0.62)	14.60 (1.15)	19.57 (4.16)	14.56 (0.97)	14.19 (0.44)	0.549	0.061
*NAFLD, %*	6.98 (1.71)	4.40 (1.37)	8.40 (2.28)	9.02 (2.44)	6.59 (1.82)	4.78 (1.84)	8.89 (2.37)	10.45 (2.85)	10.71 (2.55)	8.39 (2.45)	7.74 (0.69)	0.438	0.092
16–18 y	*n* = 266	*n* = 183	*n* = 187	*n* = 165	*n* = 199	*n* = 144	*n* = 167	*n* = 168	*n* = 158	*n* = 141	*n* = 1778		
BMI SDS	−0.17 (0.08)	−0.14 (0.12)	−0.12 (0.11)	−0.08 (0.10)	0.08 (0.11)	0.16 (0.13)	0.19 (0.13)	−0.16 (0.13)	0.08 (0.12)	0.10 (0.12)	−0.01 (0.04)	0.139	**0.023**
*BMI percentile*												0.753	**0.034**
*Normal, %*	85.06 (2.51)	82.09 (3.07)	81.04 (3.16)	80.25 (3.58)	76.34 (3.43)	76.48 (4.02)	75.65 (3.49)	79.84 (3.21)	77.76 (3.55)	77.87 (3.58)	79.35 (1.06)		
*Overweight, %*	8.44 (1.83)	6.17 (2.03)	7.27 (2.37)	9.87 (2.49)	11.09 (2.46)	8.41 (2.51)	8.39 (2.25)	7.77 (2.12)	7.57 (2.13)	10.75 (2.67)	8.54 (0.73)		
*Obesity, %*	6.50 (1.61)	11.74 (2.74)	11.69 (2.49)	9.88 (2.45)	12.57 (2.51)	15.11 (3.27)	15.97 (2.93)	12.40 (2.87)	14.67 (2.93)	11.38 (2.68)	12.11 (0.84)		
WC, cm	71.57 (0.57)	72.13 (0.89)	72.31 (0.76)	71.78 (0.70)	71.91 (0.76)	73.33 (0.89)	74.70 (0.80)	73.15 (0.87)	73.47 (0.82)	73.42 (0.91)	72.74 (0.25)	0.076	**<0.01**
*Abdominal obesity, %*	6.49 (1.62)	9.11 (2.51)	12.47 (2.58)	9.18 (2.47)	8.87 (2.17)	14.34 (2.98)	15.80 (2.93)	15.54 (2.97)	13.84 (2.93)	11.71 (2.66)	11.69 (0.83)	0.136	**<0.01**
Glucose, mg/dL	87.08 (0.46)	87.46 (0.54)	86.85 (0.59)	86.92 (0.55)	88.54 (0.51)	89.93 (0.69)	90.21 (0.71)	90.25 (0.55)	89.29 (0.67)	90.14 (0.51)	88.61 (0.19)	**<0.01**	**<0.01**
*Prediabetes, %*	3.50 (1.51)	2.83 (1.52)	4.02 (1.71)	0.68 (0.39)	5.12 (1.90)	8.24 (2.40)	7.01 (2.31)	9.64 (2.57)	7.65 (2.86)	5.68 (2.00)	5.38 (0.65)	**0.028**	**<0.01**
AST, IU/L	17.20 (0.32)	17.64 (0.52)	17.93 (0.55)	18.93 (0.96)	17.52 (0.49)	17.51 (0.64)	18.17 (0.48)	17.96 (0.62)	17.90 (0.47)	19.10 (0.79)	17.98 (0.19)	0.449	0.138
ALT, IU/L	14.20 (0.51)	15.05 (1.07)	16.57 (1.82)	16.11 (1.28)	16.67 (1.05)	15.92 (1.21)	16.32 (1.16)	16.15 (1.29)	16.06 (1.16)	19.51 (2.48)	16.23 (0.44)	0.212	0.072
*NAFLD, %*	8.82 (1.85)	8.11 (2.24)	11.33 (2.54)	10.94 (2.84)	13.07 (2.62)	14.77 (3.69)	11.93 (2.76)	15.04 (3.23)	13.66 (2.98)	13.54 (3.07)	12.08 (0.89)	0.691	**0.026**

Continuous variables are presented as the mean (standard error) and categorical data as the percentage (standard error) *with Italic*. Statistically significant *p* value is presented in bold type. For calculating p value, analysis of variance was used to compare the mean values of the continuous variables and the Rao–Scott chi-squared test was used to compare categorical variables. Linear trend analysis was performed using coefficients of orthogonal polynomials to calculate *p* for trend. BMI: body mass index; SDS: standard deviation score; WC: waist circumference; AST: aspartate aminotransferase; ALT: alanine aminotransferase; NAFLD: non-alcoholic fatty liver disease.

**Table 3 biomedicines-10-00584-t003:** Trends of prediabetes of the participants according to BMI classification.

Variable	2009	2010	2011	2012	2013	2014	2015	2016	2017	2018	Total	*p*	*p* for Trend
Normal	*n* = 799	*n* = 619	*n* = 563	*n* = 537	*n* = 550	*n* = 373	*n* = 403	*n* = 450	*n* = 415	*n* = 390	*n* = 5099		
WC, cm	66.65 (0.34)	66.38 (0.45)	65.93 (0.36)	66.10 (0.33)	66.28 (0.35)	67.11 (0.45)	68.50 (0.42)	66.75 (0.40)	66.72 (0.43)	66.69 (0.41)	66.66 (0.12)	**<0.01**	**0.020**
*Abdominal obesity, %*	0.70 (0.37)	0.59 (0.29)	0.81 (0.43)	0.05 (0.05)	1.29 (0.65)	1.13 (0.63)	0.58 (0.33)	0.85 (0.41)	0.59 (0.58)	0.47 (0.36)	0.70 (0.14)	0.696	0.993
Glucose, mg/dL	88.71 (0.26)	88.26 (0.32)	88.32 (0.39)	88.86 (0.38)	90.22 (0.38)	91.11 (0.43)	90.80 (0.32)	91.07 (0.34)	91.04 (0.42)	91.35 (0.37)	89.85 (0.12)	**<0.01**	**<0.01**
*Prediabetes, %*	4.20 (0.87)	3.45 (0.89)	6.25 (1.39)	2.65 (0.69)	7.61 (1.31)	10.13 (1.60)	7.49 (1.41)	9.05 (1.56)	10.80 (1.96)	9.87 (1.46)	6.91 (0.43)	**<0.01**	**<0.01**
AST, IU/L	18.37 (0.20)	18.32 (0.28)	18.14 (0.24)	18.51 (0.35)	17.67 (0.20)	18.67 (0.38)	19.00 (0.30)	18.44 (0.35)	19.58 (0.63)	19.41 (0.38)	18.58 (0.11)	**<0.01**	**<0.01**
ALT, IU/L	12.86 (0.26)	12.51 (0.41)	12.10 (0.32)	12.48 (0.35)	12.46 (0.27)	13.33 (0.58)	13.06 (0.43)	12.88 (0.65)	14.69 (1.58)	13.46 (0.62)	12.94 (0.20)	0.327	**0.038**
*NAFLD, %*	4.58 (0.83)	2.94 (0.85)	3.03 (0.89)	4.24 (1.16)	3.35 (0.75)	6.12 (1.45)	5.23 (1.15)	7.70 (1.70)	5.24 (1.25)	4.57 (1.13)	4.61 (0.36)	0.068	**0.019**
Overweight	*n* = 99	*n* = 81	*n* = 62	*n* = 53	*n* = 68	*n* = 45	*n* = 56	*n* = 50	*n* = 50	*n* = 44	*n* = 608		
WC, cm	78.06 (0.63)	78.64 (0.70)	76.82 (0.98)	80.11 (0.83)	76.68 (0.97)	78.23 (0.74)	79.63 (0.86)	82.16 (0.99)	76.58 (0.84)	79.12 (1.26)	78.58 (0.29)	**<0.01**	0.158
*Abdominal obesity, %*	16.67 (4.25)	19.89 (5.36)	18.82 (6.17)	13.91 (5.65)	9.07 (3.47)	12.18 (5.27)	25.21 (5.88)	30.68 (6.72)	9.90 (4.67)	18.38 (7.68)	17.45 (1.78)	0.181	0.669
Glucose, mg/dL	89.02 (0.82)	91.16 (1.07)	89.27 (1.17)	89.06 (1.11)	91.98 (1.06)	91.53 (1.24)	92.51 (0.90)	92.07 (0.96)	90.82 (1.24)	91.47 (1.05)	90.79 (0.34)	0.060	**0.019**
*Prediabetes, %*	8.12 (3.23)	8.78 (3.03)	11.22 (5.14)	3.29 (2.01)	11.64 (4.09)	11.68 (5.19)	9.34 (4.03)	13.99 (5.35)	11.38 (4.72)	6.80 (3.33)	9.55 (1.30)	0.800	0.502
AST, IU/L	19.42 (0.91)	20.91 (1.18)	20.39 (1.37)	22.97 (2.23)	20.50 (1.03)	17.49 (0.87)	19.74 (1.30)	19.00 (0.79)	18.21 (1.31)	24.63 (2.28)	20.31 (0.46)	**0.048**	0.919
ALT, IU/L	20.17 (2.49)	21.66 (2.76)	21.58 (3.30)	27.28 (5.88)	22.06 (2.43)	14.87 (1.54)	20.36 (3.07)	18.81 (1.79)	16.33 (2.69)	32.88 (9.55)	21.53 (1.28)	0.072	0.813
*NAFLD, %*	18.99 (4.61)	16.81 (4.57)	17.38 (4.97)	26.12 (7.11)	21.92 (5.82)	5.37 (3.00)	19.26 (5.81)	22.01 (6.93)	6.89 (3.83)	27.78 (7.89)	18.42 (1.84)	0.133	0.907
Obesity	*n* = 67	*n* = 71	*n* = 74	*n* = 53	*n* = 54	*n* = 48	*n* = 76	*n* = 69	*n* = 52	*n* = 56	*n* = 620		
WC, cm	85.35 (1.05)	85.86 (1.03)	87.79 (1.14)	86.30 (1.16)	88.71 (1.64)	87.61 (1.54)	89.17 (1.22)	87.76 (1.08)	88.76 (1.61)	88.86 (1.45)	87.73 (0.41)	0.236	**0.011**
*Abdominal obesity, %*	56.99 (7.05)	66.03 (7.13)	74.69 (5.93)	72.65 (6.11)	70.66 (6.53)	76.65 (7.01)	82.48 (4.14)	79.78 (5.20)	80.58 (6.70)	74.60 (7.62)	74.40 (2.01)	0.264	**0.024**
Glucose, mg/dL	90.64 (1.10)	90.94 (1.38)	90.11 (0.88)	88.64 (1.32)	90.70 (0.91)	93.51 (1.06)	93.90 (1.42)	93.45 (1.11)	91.78 (0.96)	93.05 (1.20)	91.75 (0.38)	**0.019**	**<0.01**
*Prediabetes, %*	12.56 (4.48)	12.19 (4.73)	5.30 (3.01)	6.24 (3.09)	9.22 (4.51)	13.44 (5.38)	18.19 (5.05)	18.55 (5.39)	9.28 (4.30)	17.01 (5.29)	12.26 (1.50)	0.302	0.078
AST, IU/L	21.44 (1.42)	21.07 (1.11)	23.75 (1.70)	24.91 (3.69)	22.86 (1.80)	19.99 (1.55)	25.49 (4.04)	21.18 (1.29)	23.19 (1.51)	23.87 (1.65)	22.91 (0.75)	0.582	0.543
ALT, IU/L	26.09 (3.24)	25.09 (2.58)	34.98 (5.72)	26.92 (4.59)	31.20 (3.65)	24.18 (3.06)	32.41 (7.52)	26.47 (3.42)	31.39 (3.71)	33.37 (4.10)	29.56 (1.54)	0.478	0.338
*NAFLD, %*	37.55 (7.22)	29.59 (6.51)	44.33 (6.55)	34.37 (8.36)	47.83 (7.56)	30.53 (8.85)	37.64 (6.19)	32.29 (6.67)	53.75 (7.78)	53.14 (7.57)	40.18 (2.33)	0.160	0.093

Continuous variables are presented as the mean (standard error) and categorical data as the percentage (standard error) *with Italic*. Statistically significant *p* value is presented in bold type. For calculating p value, analysis of variance was used to compare the mean values of the continuous variables and the Rao–Scott chi-squared test was used to compare categorical variables. Linear trend analysis was performed using coefficients of orthogonal polynomials to calculate *p* for trend. BMI: body mass index; WC: waist circumference; AST: aspartate aminotransferase; ALT: alanine aminotransferase; NAFLD: non-alcoholic fatty liver disease.

**Table 4 biomedicines-10-00584-t004:** Odds ratio of prediabetes and NAFLD according to each parameter.

	Prediabetes		NAFLD	
	OR (95% CI)	*p*	OR (95% CI)	*p*
BMI percentile				
Normal	Reference	Reference	Normal	Reference
Overweight	1.42 (1.04–1.95)	0.828	4.67 (3.49–6.24)	0.103
Obesity	1.88 (1.41–2.52)	**<0.01**	13.89 (10.91–17.66)	**<0.01**
Abdominal obesity	1.85 (1.39–2.47)	**<0.01**	9.34 (7.39–11.81)	**<0.01**
Prediabetes	**-**	**-**	1.85 (1.37–2.52)	**<0.01**
NAFLD	1.85 (1.37–2.52)	**<0.01**	**-**	**-**

Logistic regression analyses were performed with prediabetes and NAFLD as dependent variables. Statistically significant *p* value is presented in bold type. NAFLD: non-alcoholic fatty liver disease; OR: odds ratio; CI: confidence interval; WC: waist circumference.

## Data Availability

The data that support the findings of this study are available from the corresponding author upon reasonable request.

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
