# Peer review of "Trends in Prediabetes and Non-Alcoholic Fatty Liver Disease Associated with Abdominal Obesity among Korean Children and Adolescents: Based on the Korea National Health and Nutrition Examination Survey between 2009 and 2018"

_biomedicines, 2022, doi:10.3390/biomedicines10030584_

Round 1

Reviewer 1 Report

The manuscript analyzed the trends in prediabetes and NAFLD associated with abdominal obesity among Korean Youth of 10-18y, and found that the prevalence of prediabetes and NAFLD was increasingly associated with abdominal obesity, this study provides meaningful evidence for the prevention and treatment of related diseases.

Some minor issues should be concerned:

  1. The United Nations, for statistical purposes, defines ‘youth’, as those persons between the ages of 15 and 24 years, without prejudice to other definitions by Member States (Secretary-General’s Report to the General Assembly, A/40/256, 1985). The authors should explain why Korean youth are between 10-18 y.
  2. Line 110-111, Can only one ALT levels be used as diagnosing NAFLD?
  3. Line 169-172, What is the meaning of “Height SDS, Weight SDS, BMI SDS” in Table 1? Many readers may not understand them.
  4. Line 132-134, Is there some data of “generalized and abdominal obesity” in Figure 2 (Line 190-194)?

Author Response

The manuscript analyzed the trends in prediabetes and NAFLD associated with abdominal obesity among Korean Youth of 10-18y, and found that the prevalence of prediabetes and NAFLD was increasingly associated with abdominal obesity, this study provides meaningful evidence for the prevention and treatment of related diseases.

First of all, we are delighted that you had a favorable interest in our findings and positive assessment of our study. Also, we are very thankful for the important comments and thoughtful suggestion. For better appearance and broad readership, we revised new and clarifying statements in all parts of the manuscript as described in red font. We hope that you agree with our revision.

Some minor issues should be concerned:

  1. The United Nations, for statistical purposes, defines ‘youth’, as those persons between the ages of 15 and 24 years, without prejudice to other definitions by Member States (Secretary-General’s Report to the General Assembly, A/40/256, 1985). The authors should explain why Korean youth are between 10-18 y.

Thank you for your sharp comment. We apologize for imprecise definition of ‘youth’. However, the method of investigation is somewhat different between the subjects with age below 18 and those with age above 18 in Korea National Health and Nutrition Examination Survery. In addition, the laboratory data is investigated among the subjects with age above 10 years. Moreover, ‘youth’ is defined as those persons between aged below 18 years legally in Korea. Nevertheless, we agree that ‘youth’ should be defined as those persons between the ages of 15 and 24 years. Therefore, we clarify the age of subjects through changing the term ‘youth’ to ‘children and adolescents’ in this article.

  1. Line 110-111, Can only one ALT levels be used as diagnosing NAFLD?

Thank you for your important comment. The gold standard of NAFLD definition is biopsy and imaging studies are recommended as non-invasive diagnosis. However, biopsy is very invasive and imaging study is expensive. Thus, NAFLD can be defined as ALT elevation without other causes of liver disease in children and adolescents, especially in population-based studies. Korea National Health and Nutrition Examination Survey does not include data of liver biopsy or imaging studies. Thus, we defined NAFLD as elevated ALT (> 26 U/L for boys and > 22 U/L for girls) without hepatitis B virus or hepatitis C virus infection (references: Gastroenterology 2010, 138, 1357-1364, 1364.e1351-1352. / Int. J. Environ. Res. Public Health 2017, 14(5), 46 / J Pediatr Gastroenterol Nutr. 2018 Jul;67(1):75-79 / J Pediatr. 2021 Nov 20;S0022-3476(21)01114-8). We provided the limitation in the discussion as below:

Ultrasonography and liver biopsy were not performed to diagnose NAFLD be-cause KNHANES does not include information on ultrasonography and liver biopsy.

  1. Line 169-172, What is the meaning of “Height SDS, Weight SDS, BMI SDS” in Table 1? Many readers may not understand them.

Thank you for your sharp comment. SDS is abbreviation of ‘standard deviation score’. We agree that the readers can be confused about the definition of SDS. We provided the sentence about SDS in the method as below:

Height, weight, and BMI were presented as standard deviation score (SDS) values accord-ing to sex and age based on the 2017 Korean National Growth Charts

  1. Line 132-134, Is there some data of “generalized and abdominal obesity” in Figure 2 (Line 190-194)?

Thank you for your sharp comment. We agree that the readers can be confused with this sentence. We corrected the sentence as below:

From 2009 to 2018, the proportion of prediabetes and NAFLD increased (Figure 2), as well as those of generalized and abdominal obesity increased.

Reviewer 2 Report

The authors analyzed the trends of prediabetes and NAFLD among the youth in Korean. This is an interesting topic, but the authors need to address the following points to improve their manuscript:

Abstract: the authors did not mention anything about the abdominal obesity, but it appeared in the conclusion statement, which is confusing and abrupt. Please restructure your abstract.

Line 111: The SAFETY study had a narrower age range than your study, making me concern that your NALFD cutoff was not precise. More references need to be provided to support your methodology.

Use p < 0.01 wherever necessary, since p < 0.001 does not provide any extra information (i.e., Line 137, 140, Figure 2)

Table 1-3: In the figure legend, specify which variables are presented as continuous and which are categorical. Bolden the data that are statistically significant.  

Section 3.4. Please provide more detailed explanation for your ORs. For example, the overweight population had 0.42 more odds of being prediabetic compared to the normal weight population. A simple display of ORs is repetitive with Table 4, so you can highlight some information in the main text.

Figure 3: Label the prevalence on the top of each bar. Use star as the indicator of p < 0.01.

Line 335: The key information of Ref 35 is that, macronutrient composition in the diet can impact the incidence of these chronic diseases without affecting body weight. The author should add such information to the discussion section.

Overall, the discussion section is over lengthy so that it impairs your logic flow, and some important information were lost in the words. For example, you can provide the potential explanation for the “worse trends of prediabetes and NAFLD in the normal weight participants than the obese ones” when you first mention this. The authors compared their findings to the data in other countries in terms of NAFLD, prediabetes, overweight/obesity rate. That’s too much information, and why did you select these countries?

What is the public health-related significance of exploring the association between prediabetes and NAFLD?

Author Response

The authors analyzed the trends of prediabetes and NAFLD among the youth in Korean. This is an interesting topic, but the authors need to address the following points to improve their manuscript:

We are very grateful for the interest and the thoughtful questions from the reviewer to our manuscript. We feel that the changes included in the revised manuscript in response to those recommendations have made it more reliable, and we are grateful for the encouragement to include these analyses. We hope that the editor and reviewer agree. In summary, for better appearance and broad readership, we revised new and clarifying statements in all parts of the manuscript as described in red font. In the following, we respond point-by-point to the editor and reviewers’ comments.

Abstract: the authors did not mention anything about the abdominal obesity, but it appeared in the conclusion statement, which is confusing and abrupt. Please restructure your abstract.

Thank you for your sharp comment. We agree that providing information about abdominal obesity associated with adverse trend of prediabetes and NAFLD is required to clarify the topic. We reconstructed the abstract as your comment.

Line 111: The SAFETY study had a narrower age range than your study, making me concern that your NALFD cutoff was not precise. More references need to be provided to support your methodology.

Thank you for your sharp comment. We agree about your comment one hundred percent. The subjects of SAFETY study were between the ages of 2 through 17 years which is different from the ages of our subjects. Actually, we discussed about the definition of NAFLD at the beginning of this study. SAFETY study is a representative and multiracial study which investigated ALT reference of children. Moreover, this study has been quoted in other articles which investigated NAFLD in children with various age rang (7-18 years old in ‘Int. J. Environ. Res. Public Health 2017, 14(5), 46’ / 12-18 years old in ‘J Pediatr Gastroenterol Nutr. 2018 Jul;67(1):75-79’ / 10-18 years old in ‘J Pediatr. 2021 Nov 20;S0022-3476(21)01114-8’). We provided these additional references in our manuscript.

Use p < 0.01 wherever necessary, since p < 0.001 does not provide any extra information (i.e., Line 137, 140, Figure 2)

Thank you for your valuable comment. We agree that p < 0.001 does not provide any extra information compared to p < 0.01. We changed ‘< 0.01’ to ‘p < 0.001’ in the manuscript including all table and figures.

Table 1-3: In the figure legend, specify which variables are presented as continuous and which are categorical. Bolden the data that are statistically significant.  

Thank you for your sharp comment. We provided the units for continuous variables and ‘%’ for categorical variables in the table. In addition, we provided the categorical variables in Italic font and bolden the data that are statistically significant in the table as your comment.

Section 3.4. Please provide more detailed explanation for your ORs. For example, the overweight population had 0.42 more odds of being prediabetic compared to the normal weight population. A simple display of ORs is repetitive with Table 4, so you can highlight some information in the main text.

Thank you for your valuable comment. We provided the sentence with more detailed explanation for ORs in the results as below:

In logistic regression analyses, the overweight population had 0.42 and 3.67 more odds of being prediabetic and NAFLD, respectively, compared to the normal weight pop-ulation. Odds ratios (ORs) of generalized and abdominal obesity for prediabetes were 1.88 (95% CI, 1.41–2.52; p < 0.01) and 1.85 (95% CI, 1.39–2.47; p < 0.01), and the corresponding values for NAFLD were 13.89 (95% CI, 10.91–17.66; p < 0.01) and 9.34 (95% CI, 7.39–11.81; p < 0.01), respectively (Table 4). The participants with prediabetes had 0.85 more odds of being NAFLD compared to those with normal glucose levels (95% CI, 1.37–2.52; p < 0.01).

Figure 3: Label the prevalence on the top of each bar. Use star as the indicator of p < 0.01.

Thank you for your important comment. We labeled the prevalence on the top of each bar in Figure 3. In addition, we used star as the indicator of p < 0.01 as your comment.

Line 335: The key information of Ref 35 is that, macronutrient composition in the diet can impact the incidence of these chronic diseases without affecting body weight. The author should add such information to the discussion section.

Thank you for your sharp comment. We agree that the information about the association between macronutrient composition in the diet and chronic diseases is important. We provided the information and reconstructed the discussion as below:.

The increased prevalence of participants with prediabetes and NAFLD in children and adolescents without abdominal obesity may be related to an unhealthy diet [38, 39]. High calories, carbohydrate, and fat increase the prevalence of chronic diseases associated with insulin resistance, such as NAFLD [25, 38, 40]. The increased consumption of sweetened beverages, particularly among children and adolescents, leads to increased fructose ad-sorption with increased lipogenesis, triglycerides accumulation, and liver damage due to its metabolism [41]. Moreover, macronutrient composition in the diet can impact the inci-dence of these chronic diseases without affecting body weight [38].

Overall, the discussion section is over lengthy so that it impairs your logic flow, and some important information were lost in the words. For example, you can provide the potential explanation for the “worse trends of prediabetes and NAFLD in the normal weight participants than the obese ones” when you first mention this.

Thank you for your sharp comment. We apologize for overly long discussion which lost some important information. We re-organized the discussion concisely and provided important information. In addition, we provided the potential explanation for the “worse trends of prediabetes and NAFLD in the normal weight participants than the obese ones” in the discussion as below:

Macronutrient composition in the diet can impact the incidence of these chronic diseases without affecting body weight [38]. Rhee et al. [42] reported that rapid change in dietary patterns, including an increase in energy intake with high-fat diet, might contribute to the increasing prevalence of diabetes in Asia. Ha et al. [43] reported that carbohydrate intake was higher than the recommended range, which was 87% in men and 91% in women in Korea, and was associated with an increased incidence of type 2 diabetes. Our previous study reported that daily calorie and fat intake increased with an increase in impaired fasting glucose among Korean children and adolescents [36].

The authors compared their findings to the data in other countries in terms of NAFLD, prediabetes, overweight/obesity rate. That’s too much information, and why did you select these countries?

Thank you for your important comment. We intended to show the examples of representative countries in global, western countries, and Asia. However, we agree that the information was too much. We re-organized the discussion briefly to avoid redundancy as your comment.

What is the public health-related significance of exploring the association between prediabetes and NAFLD?

Thank you for your valuable comment. The authors agree that providing public health-related significance of exploring the association between prediabetes and NAFLD is important in this article. We demonstrated the public health-related significance of exploring the association between prediabetes and NAFLD through providing the following sentences in the discussion and conclusion as your comment.

“Therefore, close monitoring of glucose levels in children and adolescents with NAFLD as well as an attentive screening of NAFLD in those with prediabetes are required to reduce future risks of cardiovascular disease.”

“Our study demonstrated the adverse trends of prediabetes and NAFLD, as well as their relationship, among Korean children and adolescents. Moreover, the adverse trend was more apparent in children and adolescents with normal BMI and abdominal obesity, even in the young age group. These findings suggest that future risks for cardiovascular and chronic liver diseases may increase. Therefore, close monitoring of levels of fasting glucose, ALT, and WC is required not only for children and adolescents who are over-weight or obese but also for those with normal BMI in the younger age group.”

Reviewer 3 Report

The paper “Trends in Prediabetes and Non-alcoholic Fatty Liver Disease associated with Abdominal Obesity among Korean Youth: Based on the Korea National Health and Nutrition Examination Survey between 2009 and 2018" by Song et al. is a study with the aim of assessing the prevalence of prediabetes and NAFLD among youths aged 10–18 years according to age, sex, and body mass index using a nationally representative survey.

The article is well written. The study has a good design. The article is logically divided into sections and subsections. The references cited are relevant and adequate. The work has an average degree of novelty and of good interest to the readers.

Comments:

  • Introduction should be improved, in particular, it should be added that “NAFLD and metabolic disorders such as prediabetes share insulin resistance as a common pathophysiological mechanism, which has led to the consideration of a more appropriate term MAFLD (metabolic associated fatty liver disease) to replace NAFLD” (DOI: 10.3390/pr9010135).
  • Line 48: it would be important to underline the increased prevalence of NAFLD in obese and diabetic patients “NAFLD prevalence peaks to the 70–90% among obese and diabetics” (DOI: 10.3390/antiox10020270).
  • Line 335-336: The increased consumption of sweetened beverages, In particular among youngers, leads to an increased fructose adsorption with an increased lipogenesis, triglycerides accumulation and liver damage due to its metabolism (DOI: 10.31083/J.RCM2203082).

Author Response

The paper “Trends in Prediabetes and Non-alcoholic Fatty Liver Disease associated with Abdominal Obesity among Korean Youth: Based on the Korea National Health and Nutrition Examination Survey between 2009 and 2018" by Song et al. is a study with the aim of assessing the prevalence of prediabetes and NAFLD among youths aged 10–18 years according to age, sex, and body mass index using a nationally representative survey.

The article is well written. The study has a good design. The article is logically divided into sections and subsections. The references cited are relevant and adequate. The work has an average degree of novelty and of good interest to the readers.

Before addressing your comment, the authors would like to express appreciation for your efforts in reviewing our manuscript. We are delighted for your interest in our manuscript and positive assessment of our work. In summary, for better appearance and broad readership, we revised new and clarifying statements in all parts of the manuscript as described in red font. In the following, we respond point-by-point to the reviewers’ comments.

Comments:

  • Introduction should be improved, in particular, it should be added that “NAFLD and metabolic disorders such as prediabetes share insulin resistance as a common pathophysiological mechanism, which has led to the consideration of a more appropriate term MAFLD (metabolic associated fatty liver disease) to replace NAFLD” (DOI: 10.3390/pr9010135).

Thank you for your important comment. The authors agree that introduction should be improved and the reference you suggested can provide more information about association between NAFLD and prediabetes. We provided the sentence and the reference as your recommendation.

  • Line 48: it would be important to underline the increased prevalence of NAFLD in obese and diabetic patients “NAFLD prevalence peaks to the 70–90% among obese and diabetics” (DOI: 10.3390/antiox10020270).

Thank you for your valuable comment. We agree that it is important to underline the increased prevalence of NAFLD in obese and diabetic patients. We provided the sentence and the reference as your comment.

  • Line 335-336: The increased consumption of sweetened beverages, In particular among youngers, leads to an increased fructose adsorption with an increased lipogenesis, triglycerides accumulation and liver damage due to its metabolism (DOI: 10.31083/J.RCM2203082).

Thank you for your important comment. information about increased consumption of sweetened beverages with related mechanism will let the readers understand the cause of increased prevalence of prediabetes NAFLD among children. We provided the sentence and the reference as your recommendation.

Round 2

Reviewer 2 Report

The authors have successfully addressed my questions and comments. The manuscript has been substantially improved. I appreciate their time and effort.

Author Response

We are very grateful for acceptance of our revised manuscript. We feel that the changes included in the revised manuscript in response to your recommendations have made it more reliable.
